# Sensory Evaluation of Common Ice Plant (*Mesembryanthemum crystallinum* L.) in Response to Sodium Chloride Concentration in Hydroponic Nutrient Solution

**DOI:** 10.3390/foods11182790

**Published:** 2022-09-09

**Authors:** Jiaqi Xia, Neil Mattson, Alina Stelick, Robin Dando

**Affiliations:** 1School of Integrative Plant Science, Cornell University, Ithaca, NY 14850, USA; 2Department of Food Science, Cornell University, Ithaca, NY 14850, USA

**Keywords:** common ice plant, leafy green, sodium chloride, sensory evaluation, hydroponics, controlled environment agriculture (CEA)

## Abstract

Common ice plant (*Mesembryanthemum crystallinum* L.) is a novel edible plant with a succulent and savory flavor. The plants display prominent epidermal bladder cells (EBCs) on the surface of the leaves that store water and sodium chloride (NaCl). The plants have high nutritional value and are adapted to saline soils. Previous research has determined the impact of NaCl on the growth and mineral content of ice plant, but as NaCl has an impact on a food’s sensory properties, an interesting question is whether saline growth media can affect the plant’s taste and texture, and if this alters consumers’ sensory response to ice plant. The objective of this study was to evaluate the sensory aspects of ice plant, as well as consumer liking in response to increasing NaCl concentration in hydroponic nutrient solution. Four-week-old seedlings of ice plant were transplanted into deep water culture (DWC) hydroponic systems and treated with five NaCl concentrations (0 M [control], 0.05 M, 0.10 M, 0.20 M, and 0.40 M NaCl). Eight-week-old plants (after four weeks of NaCl treatment) were harvested, and the middle leaves of each plant were sampled for consumer testing. A total of 115 participants evaluated various flavor, texture, and appearance aspects of ice plant and provided their liking ratings. The consumers were able to discriminate differences in salt intensity from the plants based on NaCl treatment in the hydroponic nutrient solution. Low NaCl concentrations (0.05–0.10 M) did not have obvious adverse effect on consumer liking, which aligns with the result of previous research that 0.05–0.10 M NaCl could largely stimulate the growth of ice plant. NaCl concentrations higher than 0.20 M are not recommended from both a production and consumer perspective. With increased NaCl level in plant samples, the consumers detected more saltiness, sourness, and fishiness, less green flavor, and similar levels of bitterness and sweetness. NaCl treatment had no effects on leaf appearance and texture, and the consumers’ overall liking was mainly determined by flavor. Overall, ice plant presents some unique attributes (salty and juicy) compared to other edible salad greens; however, consumer awareness of ice plant is very low, and purchase intent is relatively low as well. Consumers picture ice plant being used mainly in salads and in restaurants.

## 1. Introduction

Common ice plant (*Mesembryanthemum crystallinum* L.) is a succulent edible plant that is emerging as a new ingredient for salad. Ice plant has a high nutritional value for humans due to its abundant antioxidant compounds such as phenolics [1]. Ice plant is used as a nutraceutical, in therapeutic cosmetics, and as food [2]. Food diversity is an important factor of household food security and is correlated with peoples’ well-being [3]. Salads and raw vegetables are a critical part of U.S. diets and can effectively increase the intake of micronutrients [4]. Ice plant is a novel and high-value salad ingredient that increases the likelihood of meeting daily nutrient requirements, as well as being an addition to food diversity. As hydroponics and controlled environment technologies become more widely used to produce fresh and high-quality vegetables, greenhouse growers are looking to expand the crops they produce, and some have added ice plant into their production [5,6,7,8,9].

Sodium chloride (NaCl) has a significant impact on cultivated crops, especially in closed hydroponic systems where NaCl could accumulate over time. Neocleous and Savvas [10] observed that increasing NaCl in a closed-loop hydroponic system negatively affected zucchini’s uptake of N, K, Ca, and Mg, and >0.003 M of NaCl could cause yield loss. Hydroponic lettuce treated with a low NaCl concentration (EC = 3.7 dS·m^−1^) lost 20% of its FW, and that treated with a higher NaCl concentration (EC = 5.6 dS·m^−1^) lost 40% of its FW; K and Mg absorption was negatively affected as well [11]. Salt tolerance is also an important consideration in plant breeding. Khattabi et al. [12] tested salt tolerance of six barley varieties and found that NaCl resulted in a decreased level of proline, K, and K/Na, and an increased level of Na in two varieties that are considered salt-sensitive. In contrast, two other varieties that are salt-tolerant had a higher level of proline, K, and K/Na, and a lower level of Na. Ice plant is known for its ability to take up NaCl, and stores water and NaCl in the epidermal bladder cells (EBCs) [13]. This characteristic makes ice plant tolerant to high salinity, and, in fact, some level of NaCl is required for the optimal yield of ice plant. Research conducted by Agarie et al. [13] showed that 0.10 to 0.20 M of NaCl added to hydroponic nutrient solution caused the highest dry weight (DW) accumulation of ice plant. However, their work used older ice plant at the reproductive stage. For intended use as an edible salad green, 0.05 to 0.10 M of NaCl gave the optimum plant performance when harvested 8 weeks after seeding (3 weeks after NaCl treatments) [14]. The storage of water and NaCl in EBCs also results in an appealing salty and savory flavor, which current salad vegetables do not possess. Many salad dressings that are used to enhance flavor contain a lot of fat and calories. Mixing salty ice plant with other vegetables may be a good strategy to reduce the use of dressings and lower the risk of obesity. Additionally, adding NaCl to the nutrient solution during the production of ice plant can increase the nutritional value of this crop. An increased concentration of sodium chloride is reported to increase the accumulation of pinitol and ononitol, compounds that promote human health, with a maximum accumulation at 0.40 M of NaCl concentration [15]. However, high leaf Na accumulation was also found, for example, 100 g of fresh ice plant at 0.05 M of NaCl had 397 mg of Na vs. 0.4 M of NaCl had 1679 mg of Na [14].

Beyond optimizing yield, it is important to understand how production practices influence the consumer’s sensory acceptance of fresh vegetables. Sensory evaluation is an effective way to screen different types of food products and drop poor products from further testing [16,17]. It is also used for comparing plant cultivars and growing treatments or systems. For example, sensory evaluation methods were adopted into lettuce [18] and strawberry [19] cultivar selection programs and were even used for selecting leafy greens for a pick-and-eat scenario on the International Space Station [20]. Talavera-Bianchi [21] used sensory analysis to determine if organically grown pac choi and tomatoes had an impact on taste vs. conventionally grown counterparts. Zhao et al. [22] also conducted a sensory comparison of multiple fruits and vegetables grown organically vs. conventionally. Highly trained panelists and common consumers were hired in Talavera Bianchi’s and Zhao et al.’s studies, respectively. Both studies drew the conclusion that organically grown vegetables did not differ from conventionally grown vegetables in the sensory sense. There was an interesting finding that participants did not perceive sensory differences in a blind test but provided a higher score to samples labeled organic in the informed test, and to samples labeled organic but which were actually grown conventionally in the inverted test [23], demonstrating the importance of labeling on consumer perception and the development of expectations [24]. Currently, no research is available on consumer responses to edible ice plants, nor in response to different NaCl concentrations during production. Therefore, the objective of this project was to quantify sensory aspects of ice plant in response to NaCl concentration in hydroponic nutrient solution using a blind consumer test.

## 2. Materials and Methods

### 2.1. Plant Samples

Following our previously developed protocols for hydroponic ice plant production, plants were grown under 5 different NaCl treatments for consumer sensory evaluation [14]. Ice plant seeds (Baker Creek Heirloom Seeds, Mansfield, MO, USA) were started in rockwool cubes and fertigated with 21N–2.2P–16.5K of Jack’s All Purpose Fertilizer (JR Peters, Allentown, PA, USA) at a concentration of 150 mg·L^−1^ N. After four weeks, when the seedlings had four or five developed leaves and a moderate number of roots (a growth stage when they can handle high NaCl concentrations), they were transplanted into deep water culture (DWC) hydroponic reservoirs with a base fertilizer and one of the five NaCl treatments: 0, 0.05, 0.10, 0.20, and 0.40 M NaCl. Previous research showed that these NaCl treatments result in a huge growth difference of the plant in terms of plant size, plant weight, leaf surface area, etc., and result in, respectively, 13,558, 180,507, 227,984, 275,661, 305,226 mg·kg^−1^ Na in leaf tissue [14], which were separated enough for people to detect differences. The plants were grown for another four weeks in a controlled greenhouse environment with supplemental lighting providing a 16 h photoperiod. The temperature was maintained at a 19.8 ± 0.5 °C (mean ± std dev.) day temperature and a 19.0 ± 0.6 °C (mean ± std dev.) night temperature. After four weeks of NaCl treatment (eight weeks after seeding), the plants were harvested (cut at the rockwool surface level) and leaves were cut from the main stem. The leaves in the middle of the plants were sampled for consumer testing (Figure 1). The experiment was designed as a complete block design, with NaCl concentration as the primary factor under analysis. 

### 2.2. Consumer Testing

All testing procedures were approved by the Cornell Institutional Review Board for testing with human subjects (Cornell IRB Protocol# 1510005908). A total of 115 participants were recruited from Cornell University’s staff, faculty, and student population. Of these, 68% were female. The average age of the participants was 29.6 ± 11.1 (mean ± std dev.) years old. About half (52%) were White/Caucasian; 37% Asian/Pacific Islander; 3% Black/African American; 6% Hispanic/Latino; and 2% another race.

Each participant provided consent and received $10 in compensation. Each panelist was given five samples of ice plant based on the treatments of 0.0, 0.05, 0.10, 0.20, and 0.40 M NaCl, labeled with random three-digit codes, in a counterbalanced order. The panelists were asked about the appearance, flavor, texture, aftertaste, and their overall liking of each sample (Appendix A). Each sample was served monadically, and the participants were asked to cleanse their palate with water and a cracker between samples. Additionally, demographic (age, gender, ethnicity) information, purchase intent and options on salt consumption were collected. The panelists were finally asked to envision the occasions/situations and dishes in which they might consume ice plant.

### 2.3. Statistical Analysis

The data were collected using RedJade^®^ Sensory Software Suite (RedJade Sensory Solutions LLC, Martinez, CA, USA) and analyzed using JMP software (SAS Institute, Cary, NC, USA) and XLSTAT/Sensory module (XLSTAT Version 2018.5.52460, Addinsoft, Paris, France). ANOVA and Tukey’s Honest Significance Difference (HSD) Test were used to determine differences among treatments. A penalty analysis was performed to understand how saltiness affected the overall liking of the plant product. Chi-squared tests were performed to determine the relationship between NaCl concentration and texture panelist ratings (JAR scales). All analyses were performed at a 95% Level of Confidence. A Principal Component Analysis (PCA) and a Hierarchical Cluster Analysis (HCA) were performed to explore multivariate relationships in the data.

## 3. Results

### 3.1. Saltiness and Overall Liking

Increased NaCl concentration in the nutrient solution during hydroponic production resulted in a significantly increased salty taste intensity of the ice plant leaf samples (*p* < 0.0001). When salt levels reached 0.20 M and higher, the overall consumer liking of the plant decreased slightly from a rating between “Like slightly” and “Neither like nor dislike” (5.3/9) to a rating between “Neither like nor dislike” and “Dislike it slightly” (4.2/9) (Table 1). The penalty analysis showed one trouble spot (mean drop in overall liking of more than 2.5 points by over 20% of the panel), which was the 0.40 M treatment. Specifically, 81% of participants found the 0.40 M sample too salty and their overall liking dropped accordingly (Figure 2). A similar pattern presented for the 0.10 M and 0.20 M groups. With an increased salt level, the percent of panelists who thought it was too salty increased, and the percent of panelists who thought it was not salty enough decreased accordingly. For the no salt (control) group, about half of the panelists thought it was not salty enough, but that did not reduce their overall liking to a great extent. For the 0.05 M sample, the percent of panelists who thought it was not salty enough and the percent of panelists who thought it was too salty were both above 20%, but the penalty to their overall liking was not sufficient to cause attention.

### 3.2. Flavor

Bitterness and sweetness were not highly detected by the panel for ice plant (intensities all below 2.5), with differences between the five NaCl treatments not statistically significant (Table 1 and Figure 3). It is worth noting that fishiness, which is not common in many other leafy greens, presented in all samples of ice plant, and its level increased at 0.20 and 0.40 NaCl. The samples treated with a higher NaCl concentration also tasted slightly more sour and had a lower green flavor, but in general the intensities were not high. In the word description of flavor, the more frequent appearance of “salty” and “fishy” with increased NaCl level also corresponds to the quantitative ratings (Table 2). The word attributes of “leafy” “vegetal” and “grassy” were more associated with the control sample and their counts decreased with increased NaCl level. People also noted “bland” and “mild” about the control sample, but these were not frequently reported for the salt treated samples. Flavor liking was highly correlated with overall liking (Pearson correlation = 0.850, *p* < 0.0001), while appearance and texture liking exhibited a weaker relationship (0.375, *p* < 0.0001, and 0.490, *p* < 0.0001, respectively) among the five treatments, suggesting that flavor is the critical determinant of overall liking in response to the NaCl treatments.

Across all the NaCl concentrations tested, at least 70% of panelists did not report an aftertaste. Of those who did, at least half found it “acceptable” and described it as mostly “green/vegetative” at lower concentrations to “salty/fishy” at higher ones.

### 3.3. Texture

The participants’ opinions towards texture were neutral to slightly liked across all NaCl treatments (Table 1). Across all treatments, generally, the majority (61%+) found the juiciness and toughness just right, while all samples skewed “not crunchy enough” (~40%+) (Table 3). An increase in NaCl treatment concentration significantly reduced the perception of juiciness, that is, significantly more panelists (*p* < 0.0001) found the 0.40 M of NaCl sample not juicy enough compared to those who found 0.05 M of NaCl and 0.20 M of NaCl samples not juicy enough (Table 3).

### 3.4. Multivariate Data Analyses

The PCA (Figure 4) uncovered two main factors that accounted for 84% of the total data variability with factor 1 (F1) accounting for the vast majority of the variability in the data set (71.09%). As the NaCl concentration increased in the test samples, overall liking, flavor liking, sweetness and green flavor decreased, while salty, sour, fishy flavor and aftertaste as well as color assessment increased. This underscores the effect of increasing NaCl concentration, as it did not only affect the salty taste perception, but also influenced the perception of many other sensory attributes. F2 accounted for only 13.09% of data variability, and primarily differentiated between the perception of crunchy texture and appearance liking. For additional details, refer to Appendix A.

Three clusters were identified by the AHC analysis (Figure 5). Cluster 1 (C1) was the largest, accounting for 38% of observations. Subsequent analyses revealed that at least one third or more of observations within each NaCl group were included in this cluster. C1 was characterized by highest overall, each modality liking (means ranged between 6.5–6.9, 9 pt hedonic scale), and mid saltiness perception (mean 3.4, 7 pt scale). C2 accounted for 33% of observations. It mostly skewed to include lower NaCl samples, specifically, 44% 0.00 M of NaCl, 48% 0.05 M of NaCl and 35% 0.10 M of NaCl observations. C2 was marked by mid liking (overall 4.4, appearance 5.6, flavor 4.5, and texture 4.9 mean, 9 pt hedonic scale) and lower flavor intensity scores, including the lowest saltiness (2.7, 7 pt). C3 accounted for the remaining 28% of observations with higher NaCl content (57% 0.40 M of NaCl and 41% 0.20 M of NaCl). This cluster had the lowest overall and flavor liking scores (mean 3.0 and 2.8 respectively, 9 pt hedonic) and the highest saltiness intensity (5.4, 7 pt). Its texture and appearance liking scores were significantly lower than those for C1, while remaining on par with C2. This supports our previous analyses that it was the NaCl concentration that was largely responsible for differences in the perception of saltiness and subsequently affected overall liking. See Appendix A for details. 

### 3.5. Purchase/Consumption Preferences

Most (90%) of the participants were not familiar (T2B, 5 pt) with ice plants (20% were “not too familiar”, 70% were “not at all familiar”). Familiarity can affect purchase intent and consumer liking [25,26]. Given this lack of familiarity with ice plant, it is not surprising that only 23% said they would purchase (T2B, 5 pt) this food product, and 42% would not (B2B, 5 pt) purchase it. 

However, compared to regular salad greens, about half of the participants found it more or as appealing as other salad greens; 27% thought it was more appealing (T2B, 5 pt); and 21% thought it was as appealing (Mid-Point, 5 pt). When asking on what occasion or in what situation a panelist might find themselves eating this plant, the most frequent words were restaurant (appeared 39 times) and salad (42). Upon asking how they would prepare and serve the dish with ice plant, salad was the only high frequency word (92).

Regarding salt consumption, a little more than half of the panelists (53%) expressed that they did not worry about it. Additionally, given the two following scenarios: I like to try new foods that I have never tasted before vs. I order the dishes with which I am familiar to avoid disappointment and unpleasantness, most people (82.6%) selected the former.

Therefore, ice plant may fit well in the specialty product category, especially among novelty seekers. 

## 4. Discussion

In this study, the NaCl treatments in hydroponic nutrient solution that led to the highest overall liking were 0.0, 0.05, and 0.10 M of NaCl vs. the 0.20 and 0.40 M of NaCl treatments, which had lower liking ratings. Consumers reliably detected greater saltiness at every level as NaCl concentration increased, and this seemed to negatively affect flavor, and in turn overall liking. From the production perspective, the salt treatments of 0.05–0.10 M of NaCl also led to the greatest yield of ice plant [14], which corresponded with previous studies [2,13,27]. Given the result that consumer liking was not affected by increasing the NaCl levels to 0.05 M or 0.10 M (Table 1), we can conclude that treating ice plant with a low level of NaCl during production can increase yield without having adverse effects on consumer liking. On the other hand, since consumers could tell the differences in salt intensity (Table 1), producers would have the option to create ice plants in a regular and a low salt format without affecting liking, and thus, satisfy various market segments (i.e., individuals who prefer higher or lower salt intensity). 

From a health perspective, adding 0.20 M or higher concentrations of NaCl in the hydroponic nutrient solution during the growth of ice plant is not recommended. The recommended daily Na consumption is 2300 mg [28]. Previous research quantified the amount of NaCl in ice plant and 100 g of ice plant shoot FW contained 397 mg and 616 mg of Na for the 0.05 and 0.10 M of NaCl treatments, respectively [14]. However, 100 g of ice plant shoot FW contained 910 mg and 1679 mg of Na for the 0.20 and 0.40 M of NaCl treatments, respectively, which are likely too high considering Na intake from other foods during the day. Additionally, the penalty analysis showed that many panelists (>80%) thought the 0.40 M sample was too salty, and the corresponding mean drop in their overall liking was high (>2.5) (Figure 2). Moreover, increased sourness and especially fishiness and decreased green flavor in higher NaCl samples led to the same conclusion that low NaCl samples were more highly liked than high NaCl (Table 1).

No previous study has been conducted on the effect of NaCl in hydroponic production on the sensory aspects of ice plant, but studies on other vegetables and food products showed that NaCl could affect taste, texture, and consumer liking of the products. For example, NaCl application to hydroponic solution could increase the sweetness, acidity, umami, and overall liking of tomato fruits [29]. Salt addition increased the flavor, juiciness, and texture of restructured pork chops [30]. Higher salt concentration led to the increased firmness and decreased cohesiveness of Mozzarella cheese [31]. However, the effect of NaCl depends on specific product categories. Jaenke et al. [32] conducted a comprehensive review and concluded that salt content could be reduced substantially in bread and processed meat without lowering consumer acceptance, but salt reduction in other products such as cheese and soup required novel strategies to maintain consumer acceptance. In the current study, salt did have an effect on consumer liking, but the effect was not substantial. With increased NaCl treatment, overall consumer liking decreased slightly. The food industry is searching for ways to reduce salt in products without negatively affecting sensory quality or consumer acceptance [33]. Multiple ways such as simply reducing added salt, adding umami taste compounds, or odor–taste interactions have been proposed and evaluated. In the case of ice plant, lower salt is preferred by consumers. In other words, a decrease in consumer acceptance when salt is reduced should not be a concern.

Ice plant is known as a halophyte (i.e., plant adapted to saline conditions) that has succulent leaves [34]. It stores an abundant amount of water and salt in its tissues, specifically in EBCs on the leaf surface [13]. Therefore, it is known for its juicy and savory flavor. In this research, panelists thought that the juiciness of ice plant was just about right, which met their expectations, and the salt level did not affect juiciness (Table 3). Previous research showed that differences in leaf water content with NaCl treatment were statistically significant [14], but that differences were obviously not detectable from the consumer perspective.

Beyond saltiness or juiciness, another attribute of ice plant flavor was fishiness. Although further research and tissue analysis may be warranted, one speculation is that ice plant may use one of the same mechanisms as fish. In order to deal with the salt in seawater, fish accumulate trimethylamine oxide (TMAO) in the body, and microbes break down TMAO into trimethylamine (TMA), which emits a fishy smell [35]. Ice plant grown in the saline environment may also accumulate TMAO and TAM in response to high salt in their hydroponic water, but this needs thorough further investigation. Interestingly, research suggests a positive correlation between TMAO and heart disease, and the paradox that fish containing a lot of TMAO is generally accepted as cardioprotective [35,36,37]. Although this paradox has not been solved, there is speculation that other healthy components in fish such as omega-3 fatty acids may offset the negative effect of TMAO, or that many commonly eaten fish do not have a lot of TMAO [36]. If ice plant does contain TMAO, it may be a consideration in its nutritional profile. However, overall, ice plant is reported to have high nutritional value. For example, it has high level of polyols (pinitol, ononitol and myo-inositol), compounds that promote human health [15]. Pinitol constituted 71% of the soluble carbonate and 9.7% of the DW in the stressed leaves of ice plant [38]. Additionally, Agarie et al. [15] found that pinitol/ononitol content in ice plant increased with an increased level of NaCl treatment, reaching a maximum at 0.4 M of NaCl concentration. In the control (no salt treated) plants, although pinitol/ononitol content was relatively lower, myo-inositol content could reach 7.5 mg/g of fresh weight (FW). The conversion from myo-inositol to ononitol and then to pinitol [39] explains the high pinitol/ononitol and low myo-inositol content in NaCl treated plants. Ice plant is high in myo-inositol compared to other leafy greens such as collard, Romaine lettuce, and spinach whose myo-inositol contents are 0.64, 0.17, 0.08 mg/g FW, respectively [40]. Even compared with high-myo-inositol food such as dried prune, Great Northern beans, and peanut butter whose myo-inositol contents are, respectively, 4.78, 4.7, 4.4, 3.04 mg/g FW, ice plant is comparable. More information is needed to understand whether the fishiness attribute of ice plant affects consumer liking. One study with yogurt implies that fishiness may not be a problem, because, while trained panelists detected stronger fishiness in the treated yogurt, consumers provided the same ratings to the control and treated samples [41]. Future studies of ice plant could use the just about right (JAR) test and a penalty analysis to test this association. 

With the development of edible halophyte and halophyte-based agriculture (i.e., the use of salt-tolerant plants in agriculture so that salt-compromised land can be used) [42], ice plant as a high-nutritional crop and obligatory halophyte that requires some level of salt to grow [43] could be an excellent candidate. However, our study showed that 90% of the panelists were not familiar with ice plant. Regarding purchase intent, although most (82.6%) people expressed that they like to try new foods, only 24% expressed that they would buy (definitely or probably) ice plant. More plant-based research including nutrition analysis and consumer-based marketing should be conducted to understand this relationship. An example that may be relevant in popularizing ice plant is microgreens. Microgreens have been produced since the mid-1990s [44] but were still thought of as a new crop in 2010 [45]. Massive research and marketing have been performed in the past decade and microgreens are becoming a more popular food [46,47,48,49,50,51,52,53]. In this manner, ice plants may represent an interesting and novel food for a customer seeking diverse flavors in the salad category.

## 5. Conclusions

This study presents the first published consumer sensory evaluation of edible ice plant, as well as the first paper regarding the effect of NaCl in the hydroponic nutrient solution on leafy greens consumer evaluation. In general, the consumers had a fairly neutral level of liking for ice plant. With an increased NaCl level, the consumers were able to discriminate an increase in saltiness, and at higher levels their overall liking decreased slightly. However, the overall liking of plants treated with 0.05 M and 0.10 M did not differ significantly from control plants, indicating the feasibility of adding low concentrations of NaCl in hydroponic nutrient solutions during production, which optimizes plant yield. Adding 0.20 M of NaCl or higher concentrations is not recommended from both the production standpoint (decreasing yields) and the consumer perspective (decreasing liking and has potential health concerns). The appearance and texture did not vary with NaCl, but flavor, especially saltiness, sourness, and fishiness, drove differences in the consumer liking of ice plant. Overall, ice plant stands out as a unique edible salad green with attributes such as saltiness and juiciness that are quite different than other leafy greens. More research is necessary on the effect of NaCl concentrations during ice plant production on its human nutritional value. Most people are not familiar with ice plant and their purchase intent is relatively low, but they can picture consuming ice plant in salads and in restaurants. If a larger commercial hydroponic industry is to develop, more work is needed to acquaint people with ice plant, such as concerted efforts to educate patrons of restaurants, or offering samples at supermarkets. More exploration remains around halophyte-based agriculture (using plants that are salt adapted and that can also be used for human consumption or animal feed), such as different ways of using ice plant in various dishes, the development of ice plant genotypes that have better flavor characteristics and optimizing ice plant production for higher plant quality. Other work on the potential use of ice plant in nutraceuticals, specifically to determine the potential nutritional content of ice plant beyond polyols and phenolics such as mineral elements, vitamins, amino acids, or other antioxidant compounds may be beneficial.

## Figures and Tables

**Figure 1 foods-11-02790-f001:**
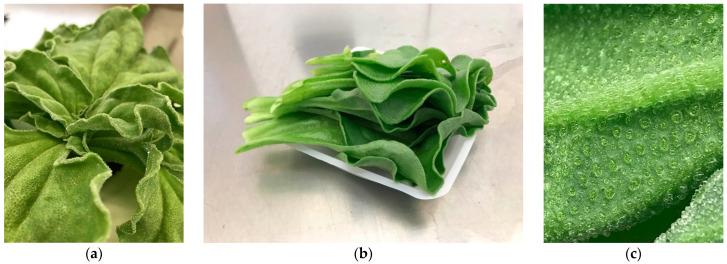
(**a**) Whole plant appearance of ice plant; (**b**) leaves harvested from one individual ice plant (but only leaves in the middle of the plants were sampled for consumer testing); (**c**) a close-up exhibition of epidermal bladder cells (EBCs) on the bottom side of the ice plant leaf.

**Figure 2 foods-11-02790-f002:**
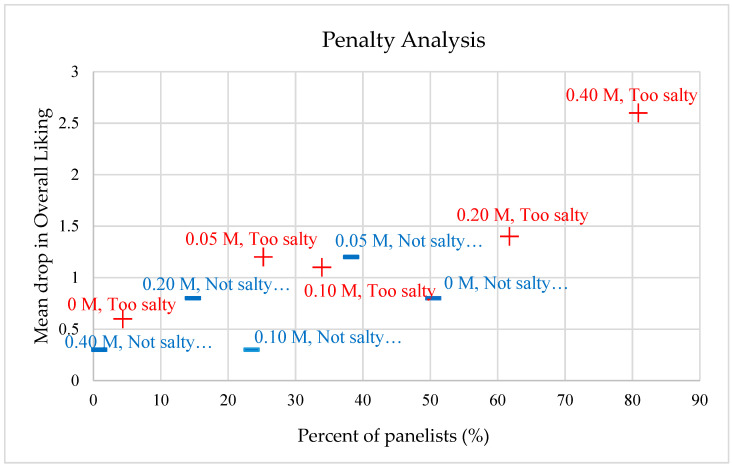
Penalty on overall liking of ice plant in response to sodium chloride (NaCl) treatment in hydroponic nutrient. The *y*-axis represents the mean difference between the sample overall liking rating, compared to the overall liking rating of panelists who found the sample “too salty” (Top 2 Box% (T2B%), 5 pt, red) or “not salty enough” (Bottom 2 Box% (B2B%), 5 pt, blue). The *x*-axis = percent panelists per category.

**Figure 3 foods-11-02790-f003:**
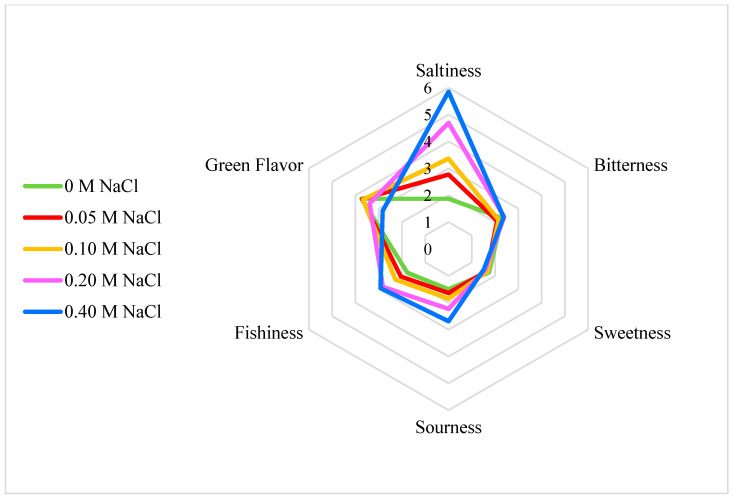
Radar plot of six flavor aspects. Polar axis represents intensity of six categories of flavor including saltiness, bitterness, sweetness, sourness, fishiness, and green flavor on a 7-point scale (1 = not at all; 7 = very strong). Data for each attribute are means of the 115 ratings.

**Figure 4 foods-11-02790-f004:**
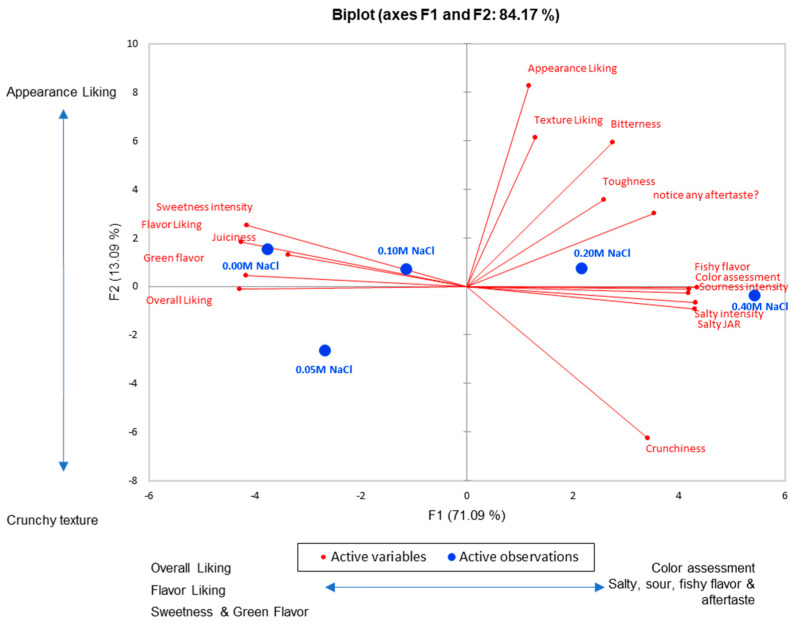
PCA Summary Plot. *x*-axis = F1 accounts for 71.09% total data variation; *y*-axis = F2—13.09% data variation. Top positive and negative factor loadings were called out for ease of interpretation.

**Figure 5 foods-11-02790-f005:**
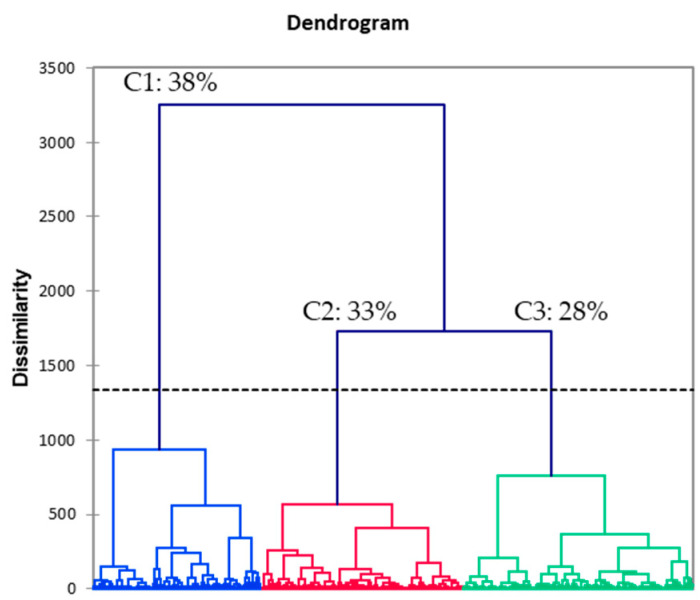
HCA dendrogram plot shows percent observation membership within each cluster (C). C1 = Cluster 1, C2 = Cluster 2, C3 = Cluster 3. Blue, red, green colors represent different cluster membership.

**Table 1 foods-11-02790-t001:** Mean rating of all hedonic (9-point scale) and flavor intensity (7-point scale) attributes of ice plant leaf samples in response to sodium chloride (NaCl) treatment in hydroponic nutrient.

	0.00 M NaCl	0.05 M NaCl	0.10 M NaCl	0.20 M NaCl	0.40 M NaCl	Pr > F (ANOVA)	Pr > F (Sample)
Overall liking (9 pt)	5.35 a	5.15 a	4.99 ab	4.43 bc	4.21 c	<0.0001	<0.0001
Appearance liking (9 pt)	6.13 a	5.86 a	6.26 a	6.35 a	6.08 a	<0.0001	0.078
Flavor liking (9 pt)	5.40 a	4.98 ab	4.99 ab	4.46 bc	4.09 c	<0.0001	<0.0001
Texture liking (9 pt)	5.84 a	5.69 a	5.75 a	5.74 a	5.84 a	<0.0001	0.937 NS
Color assessment (7 pt)	3.08 c	3.11 c	3.27 bc	3.51 b	4.29 a	<0.0001	<0.0001
Saltiness intensity (7 pt)	1.87 e	2.77 d	3.36 c	4.69 b	5.85 a	<0.0001	<0.0001
Bitterness intensity (7 pt)	2.30 a	2.09 a	2.16 a	2.39 a	2.37 a	<0.0001	0.290 NS
Sweetness intensity (7 pt)	1.74 a	1.65 a	1.67 a	1.57 a	1.52 a	<0.0001	0.306 NS
Sourness intensity (7 pt)	1.50 c	1.64 c	1.85 bc	2.24 b	2.69 a	<0.0001	<0.0001
Green flavor intensity (7 pt)	3.74 a	3.71 a	3.68 a	3.39 a	2.84 b	<0.0001	<0.0001
Fishy flavor intensity (7 pt)	1.77 c	2.05 c	2.28 bc	2.82 ab	2.92 a	<0.0001	<0.0001

Letters represent mean separation comparison across NaCl treatments using Tukey’s HSD (alpha = 0.05): same letter = not statistically different from another mean with the same letter, different letter = statistically different mean from the one with a different letter, at 95% Level of Confidence. NS = not statistically significant at 95% Level of Confidence. Data represent means (±SE) of 115 panelist rating.

**Table 2 foods-11-02790-t002:** Number of most frequently appearing words in the description of the flavor of ice plant samples in response to sodium chloride (NaCl) treatment in hydroponic nutrient solution.

0 M of NaCl		0.05 M of NaCl		0.10 M of NaCl		0.20 M of NaCl		0.40 M of NaCl	
Word	Count	Word	Count	Word	Count	Word	Count	Word	Count
watery	20	salty	32	salty	48	salty	63	salty	86
green	17	bland	15	bitter	9	fishy	20	fishy	15
bland	15	lettuce	10	fishy	9	bitter	8	sour	11
lettuce	8	vegetal	9	bland	9	sour	5	green	6
salty	8	bitter	8	green	8	bland	5	bitter	4
bitter	7	fishy	9	lettuce	8	green	4	grassy	4
vegetal	7	watery	6	vegetal	6	leafy	4	water	3
leafy	12	grassy	8	mild	5	vegetal	4	chip	2
grassy	8	green	6	neutral	5	grassy	3	leafy	2
mild	5	leafy	3	sour	5	lettuce	3	lettuce	2

**Table 3 foods-11-02790-t003:** Frequency counts (%) on 5-point just about right (JAR) scale of crunchiness, juiciness, and toughness intensity of ice plant in response to sodium chloride (NaCl) treatment in hydroponic nutrient. Same letters represent statistically significant groups that are different from each other based on k proportion test (alpha = 0.05). Bold = statistically significant at 95%; NS = not statistically significant at 95% Level of Confidence.

%	Crunchiness			Juiciness			Toughness		
	Not Crunchy	Just Right	Too Crunchy	Not Juicy	Just Right	Too Juicy	Not Tough	Just Right	Too Tough
0.00 M NaCl	56	44	0	14 AB	68	18 AB	35	63	3
0.05 M NaCl	43	55	2	10 A	72	18 AB	37	61	2
0.10 M NaCl	48	50	2	13 AB	63	23 B	30	66	4
0.20 M NaCl	43	57	0	11 A	71	17 AB	36	62	3
0.40 M NaCl	39	59	2	29 B	63	9 A	33	64	3
Χ^2^ *p*-value	0.127 NS	0.184 NS	0.400 NS	0.000	0.405 NS	0.056 NS	0.770 NS	0.929 NS	0.819 NS

## Data Availability

The data presented in this study are available on request from the corresponding author.

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
