# Peer review of "Sensory Evaluation of Common Ice Plant (Mesembryanthemum crystallinum L.) in Response to Sodium Chloride Concentration in Hydroponic Nutrient Solution"

_foods, 2022, doi:10.3390/foods11182790_

Round 1

Reviewer 1 Report

I reviewed the manuscript entitled, Sensory Evaluation of Common Ice Plant (Mesembryanthemum crystallinum L.) in Response to Sodium Chloride Concentration in Hydroponic Nutrient Solution. Authors only performed sensory analysis, which is not enough for publications. In addition, no other different types of sensory evaluation methodologies. Authors want to focus only on sensory evaluation. In this case, authors must use different sensory methods and provide the best sensory methods. Authors must perform PCA to understand the effect of salt concentration. Overall, the manuscript has less novelty and not enough experiments, poor discussion, and no supportive statistical approach (PCA or HCA).

The change in salt concentration may affect the texture of the sample. Thus, authors must conduct TPA. How did the authors finalize the concentration of salt? 

Reviewer 2 Report

Line 14

Plant ultimate taste properties?

Section 2.3

I assume the alpha level is set at 5%?

There's a Pearson correlation that is carried out on Table 1, this needs to be added.

Table 1

Rather than pearson correlation, wouldn't it make more sense if the authors attempted a mixed model ANOVA?

Figure 2

The penalty analysis is based on overall liking score? Clarify in the figure caption

Table 3

Here a chisq was attempted but no info in stats section, needs to be added. What's the significance of including the 90%? If the above analysis is set at 95%?

Have the authors considered to attempt a multivariate analysis to merge their results together to have a better picture on what's happening here? Some L-PLSR or some sort can assist the interpretation especially considering the salt level of the samples?

The paper is well written and the results were well discussed. Some minor information is missing and i'd recommend the authors to attempt/consider the aforementioned approaches.

Reviewer 3 Report

Dear authors, 

I have completed my review of your submission. This contains important results and the manuscript is well written. Some points must be addressed to improve the submission. Please find below my comments and suggestions: 

1- The manuscript must be revised carefully for typos as well as language mistakes,

2- The authors must justify the application of their treatment at 4 weeks after sowing. Likewise, the choice of such concentrations of sodium chloride has to be justified, 

3- Line 47, please add a reference, 

4- The authors must deepen the state-of-art-of research by adding newly published literature even on other species to highlight the impact of NaCl on biochemical, physiological, and nutritional levels of cultivated plants. Here some papers that must be added and discussed (https://doi.org/10.21273/HORTSCI16246-22; https://doi.org/10.33263/BRIAC123.27872799; https://doi.org/10.1111/ijfs.15916; etc),

5- Plant material must be clarified; which genotype/cultivar/veriety?. Detailed information must be provided, 

6- Likewise methods and experimental design must be detailed,

7- Table 3, outcomes must be analyzed for statistics,

8- In my opinion, principal component analysis explains better the results,

Recommendation: reconsider after major revision

Kind regards.  

Round 2

Reviewer 1 Report

Authors are now answered the some of my questions. In my opinion, this version can be accepted for publication

Reviewer 3 Report

Dear Authors, 

The manuscript was greatly improved and therefore I recommend its publication within Foods. 

Kind regards.